# Run-Time Adaptive In-Kernel BPF/XDP Solution for 5G UPF

**Thiago A. Navarro do Amaral, Raphael V. Rosa, David F. Cruz Moura**  **and Christian Esteve Rothenberg \***

School of Electrical and Computer Engineering (FEEC), University of Campinas (UNICAMP),
Campinas 13083-872, SP, Brazil; t159121@dac.unicamp.br (T.A.N.d.A.); raphaelvrosa@gmail.com (R.V.R.);
dfcmoura@gmail.com (D.F.C.M.)
**\*** Correspondence: chesteve@unicamp.br

**Abstract:** Flexibility is considered a key feature of 5G softwarization to deliver a timely response to changes in network requirements that may be caused by traffic variation, user mobility, dynamic network function chains, slice lifecycle management operations, among others. In this article, we evolve the `upf-bpf`[1] open-source project by proposing a new design to improve its flexibility by reducing the run-time adaptation time. The project proposes an in-kernel solution based on BPF and eXpress Data Path (XDP) for 5G User Plane Function (UPF) implementations. The Just-In-Time (JIT) compilation may have a huge impact on the adaptation time due to the in-kernel verification of the BPF programs at run-time. Our results show latency improvements of around 95% to inject the BPF program into the Linux kernel. Furthermore, the solution keeps the same functionalities and delivers a packet processing performance of around 10–11 Mpps using 6 cores with almost 70% of the CPU utilization in downlink/uplink directions.

**Keywords:** 5G core networks; XDP; BPF; UPF; flexible; fast packet processing; Linux

## 1. Introduction

Three paradigms are shaping the design and operations of networking infrastructures: SDN [1], NFV [2], and cloud network virtualization [3]. These enabling technologies underpin the softwarization of 5G [4] delivering a more flexible, scalable, open, and programmable approach to network design and operations. Flexibility in softwarized networks can be defined as the timely support of changes in the network requirements [5], i.e., the network ability to react against events in a short period of time in order to meet the new network requirements. The main sources of events are changes in traffic patterns, user mobility, network lease, network upgrades, failure mitigation among other 5G slice lifecycle management operations [6].

The concept of flexible networking can be illustrated with an example: imagine a final match of the Soccer World Cup subject to large traffic variation in the region due to increased user equipment concentration (UEs) around the match venue. In this case, the network should be able to respond to this event by increasing the network capacity in that specific area in order to keep the Service Level Agreements (SLAs) for all subscribers. There are multiple alternatives to handle this situation such as changing the network topology to redirect part of the traffic to other paths or scaling up/out the network functions to support the higher demand. The network can be considered flexible if such adaptations are carried out in a short period of time, without the need of drastic system upgrades.

The time spent for the system to reach the new desired state is defined as adaptation time and it is related to the network domain, use case and technology. In our previous work [7], we presented the `upf-bpf` open-source project as an approach for 5G User Plane Functions (UPF) for Multi-Access Edge Computing (MEC) environments based on BPF/XDP technologies [8]. For BPF-based [9] solutions, the JIT compilation of the BPF program may reflect the adaption time of the network due to the in-kernel verification in

run-time. The time spent to inject the program into the kernel is correlated to the size of the BPF instructions [10].

In this article, we first discuss the main encountered limitations of the `upf-bpf` approach and then propose a new design to improve the flexibility with reduced run-time adaptation time. Our results show a time improvement of 96% to inject the BPF program into the Linux kernel. Furthermore, the solution retains all functionalities at high-performance, i.e., around 10-11Mpps using 6 cores with almost 70% of the CPU utilization in downlink/uplink directions. Altogether, the main contributions of this work related to the reference design for run-time adaptive in-kernel BPF/XDP solutions for 5G UPF and the open-source implementation available in the `upf-bpf` repository (https://github.com/navarrothiago/upf-bpf, accessed on 18 March 2022).

The rest of the article is organized as follows. Section 2 provides an overview of the BPF/XDP, the 5G System (5GS) architecture, and the `upf-bpf` open-source project. Section 3 elaborates on the current limitations of the current `upf-bpf` design. Section 4 presents the new design in order to overcome the identified limitations and discusses the main differences with the previous version. Section 5 describes the setup and the test cases for the performance evaluation regarding the scalability and the adaptation time when injecting the BPF program in each version. Section 6 discusses related work. Conclusions and future work are presented in the last section.

## 2. Background

This section provides relevant background on BPF/XDP and 5G user plane functions.

### 2.1. BPF

BPF [9], well known as eBPF, is a general purpose engine that allows you to execute instructions in the Linux kernel itself with two main goals: (i) to deliver negligible overhead when mapping these instructions to native instructions, and (ii) to guarantee that the program is safe at load time. This technology has been available since kernel version 3.14.

The engine is composed of 11 64-bit registers (https://github.com/torvalds/linux/blob/v4.20/include/uapi/linux/bpf.h#L45 accessed on 18 March 2022), a program counter and 512 bytes stack. Programs can be implemented using the "strict C" language and compiled using GNU (GNU Compiler Collection) or LLVM as backend (https://git.kernel.org/pub/scm/linux/kernel/git/torvalds/linux.git/tree/Documentation/networking/filter.rst accessed on 18 March 2022) (Figure 1). After being compiled, the generated bytecode can be loaded via the $bpf$ system call using the $BPF\_PROG\_LOAD$ command. From there, two steps are performed: verification and, if enabled, JIT compilation (Just-In-Time).

The verification step is done by the Verifier. It ensures that the program runs safely, for example blocking its loading otherwise. One of the checks is against infinite loops. These loops are prohibited in BPF programs, therefore ensuring that the program is always terminated.

After verification, the bytecode can be JIT-compiled inside the kernel as a performance optimization. In this step, the transformation of the generic bytecode into machine instructions native to the hardware that the operating system is running takes place. For this reason, the BPF program can run just as efficiently as kernel modules or natively compiled code [11].

After being loaded and verified, BPF programs can be hooked in specific areas (e.g., network stack) in the Linux kernel and its execution is triggered by an event (e.g., a received packet). One of them is XDP, which will be introduced in the next section. The BPF portability means that the BPF bytecode works correctly across different kernel versions without the need to recompile it for each specific kernel version. The BPF Compile Once, Run Everywhere (CO-RE) [12] is the main technology behind this concept.

BPF programs can store different types of data in generic structures called Maps. Each data is accessed by a key. Maps can be shared between userspace programs and BPF programs. More information about BPF is available at [11,13].

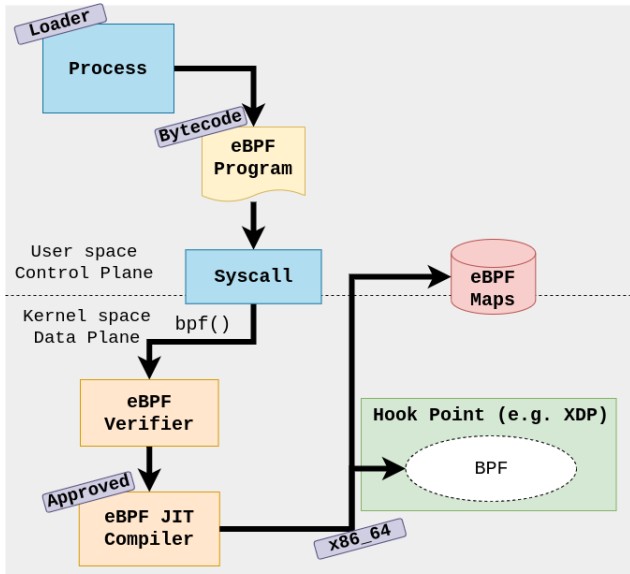

**Figure 1.** Diagram representing the BPF program load flow in the kernel. Source: adapted from [11].

One of the main advantages of using this type of technology is to enable programmability of the kernel without having to change its source code or install additional modules. Furthermore, BPF programs have a stable ABI (Application Binary Interface) that guarantees BPF programs keeps running with newer kernel versions.

On the other hand, BPF programs have restrictions that make it challenging to develop these programs to perform more complex functions. Among them, the following can be highlighted:

- Limit of 5 arguments for a function;
- Limit of 32 nested tail call calls;
- Infinite loops are not allowed;
- Send the same packet to multiple ports;
- Only 30 data structure BPF maps types available (kernel v5.16.10) (https://elixir.bootlin.com/linux/v5.16.10/source/include/uapi/linux/bpf.h#L878 accessed on 18 March 2022);

Some of these restrictions as well as workarounds for them are discussed in [10].

### 2.2. XDP

XDP [13] enables fast packet processing within the Linux kernel through a hook point located in the reception chain of a network device driver before memory (i.e., socket buffer) allocation by the operating system. When the packet is received by the network device driver, the hook which contains the BPF program is executed. The BPF program can perform different actions, such as dropping the packet, redirecting it to another interface, or sending it to be processed in the Linux network stack (Figure 2). The network device driver must support XDP to take full advantage of its benefits. If not supported, the program runs in generic mode at lower performance. It is worth mentioning that the portability for this kind of program is almost transparent to the developer due to the extra layer abstraction between the kernel and BPF context that is represented by its structures. For instance, there is a mapping between the BPF structure (xdp_md) into internal kernel structure (xdp_buff) (https://elixir.bootlin.com/linux/v5.16.10/source/net/core/filter.c#L9116 accessed on 18 March 2022).

### 2.3. Libbpf

Libbpf (https://www.kernel.org/doc/html/latest/bpf/libbpf/index.html accessed on 18 March 2022) is a userspace library for loading and interacting with BPF programs. It

is part of the Linux source tree and it is the reference library de facto. It is an alternative for the BCC (BPF Compiler Collections) toolkit. BCC has some disadvantages when comparing with libbpf, for instance, the userspace program has high footprint due to the clang compiler dependency during run-time, which uses the compiler to build the BPF program during the user program execution. This means that the development of the BPF code becomes harder, because the compilation errors are detected at run-time instead of the compilation time. So, the adoption of libbpf creates a userspace program much smaller and improves the software life cycle of the BPF program. Another advantage is that libbpf is used by BPF CO-RE technology. The `upf-bpf` project is based on libbpf.

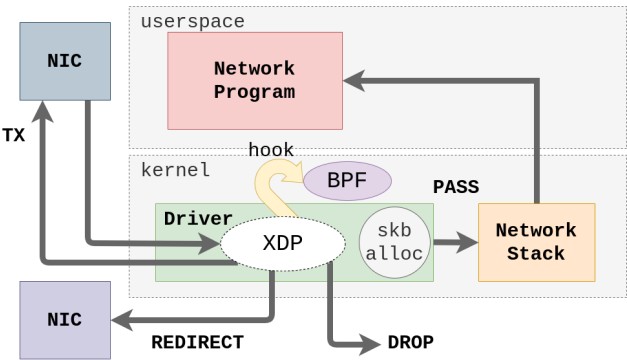

**Figure 2.** Simplified XDP architecture diagram with the possible actions that can be applied to the received packet before the *socket buffer* allocation by the operating system. Source: adapted from [14].

### 2.4. 5G Network Architecture

The architecture of 5G networks is represented in Figure 3. The colored regions represent the 5G network core, called New Generation Core (NGC), where the green block represents the User Plane (UP), while the blue ones, the Control Plane (CP). One of the main changes regarding LTE networks is to provide a Service-Oriented Architecture (SOA), i.e., composed of virtual network functions with well-defined interfaces. These functions can be performed on commodity hardware equipment and accessed through a communication protocol (e.g., HTTP). Furthermore, the 5G architecture follows the Control and User Plane Separation (CUPS) model. This paradigm enables the deployment of UP closer to applications and services, thus reducing latency and traffic in the core of these networks.

In the next sections, two relevant components of the NGC are detailed: Session Management Function (SMF) and UPF.

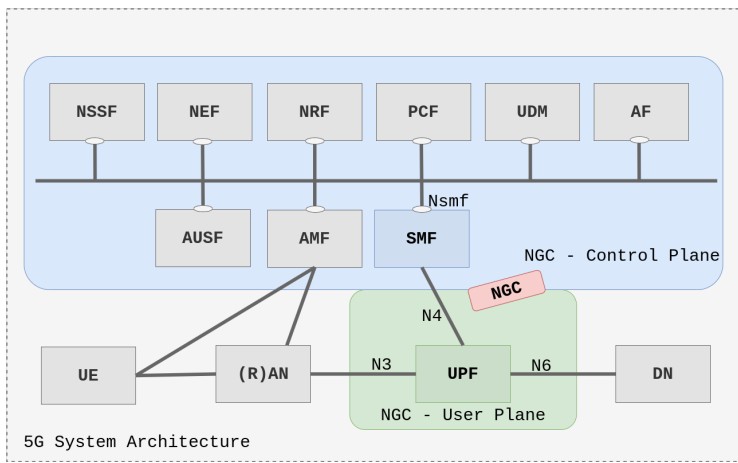

**Figure 3.** Simplified 5G System architecture. Highlighted, the SMF and UPF components are addressed in the next sessions. The colored region represents the core of the 5G network.

### 2.4.1. SMF

SMF is part of the 5G CP. One of the most relevant SMF functions is to manage the sessions used for data traffic between the UE and the Data Network (DN), namely Session Protocol Data Units (PDU). Each session is represented by a logical tunnel passing through the UE, Radio Access Network (RAN), and UPF. When a PDU session is established, a context is created in each of these components. A context contains a set of specific rules that will be applied in the data packet in order to guarantee Quality of Service (QoS) and to forward the packet to the next hop, for instance. In the case of UPF, this context is represented by the Packet Forwarding Control Protocol (PFCP) Session.

PFCP is the signaling protocol used for communication between SMF and UPF components. This communication involves PFCP session management procedures. The communication interface between the two components is represented by reference point N4 in Figure 3.

CP components communicate with SMF through the service provided by Nsmf_PDU-Session [15]. This service involves PDU session creation, update, and removal procedures. One session establishment use case example is when the UE is registered at the core of the network and has data to send to the DN. If there is no session established, the procedure for requesting session establishment will be carried out as shown in Figure 4. After performing the session establishment, the UE will be able to send/receive data in uplink/downlink direction.

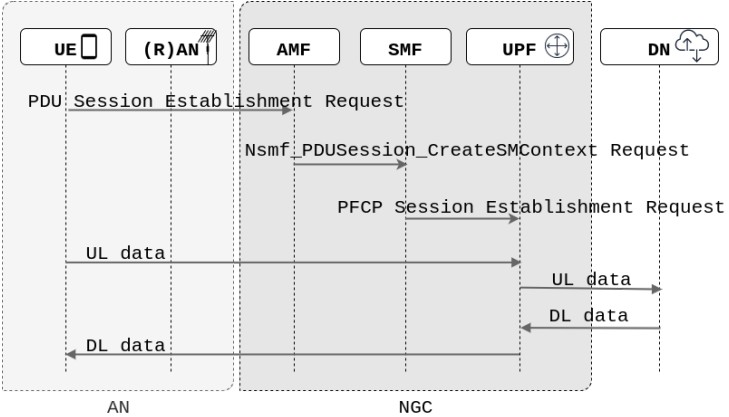

**Figure 4.** Simplified procedure for requesting session establishment.

Compared with LTE networks, the SMF encompasses a set of functionalities from the Mobility Management Entity (MME) component within Evolved Packet Core (EPC). Additionally, the N4 reference point encompasses a set of functionalities defined for the Sxa/Sxb/Sxc interfaces [16], which are also used for signaling between the CP and the UP [17].

### 2.4.2. UPF

One of the main user plane components for NGC is the UPF. It is responsible for several functionalities related to user data traffic, such as forwarding and routing packets, applying rules to ensure quality of service, generating traffic usage reports, and inspecting packets [15]. It works like a gateway between the RAN and the external DN (e.g., Internet, IP Multimedia System, local data network, etc). Regarding LTE networks, the UPF encompasses the functionalities of Servicing Gateway (SGW) and Packet Data Network Gateway (PGW) (In this work, the term SPGW will be used to refer to the combination of the SGW and PGW components as specified in [16]. SPGWu represents the component responsible for implementing the user plan functionalities for LTE networks).

Data traffic takes place through a PDU session that is represented by contexts stored in the UE, RAN, and UPF components. In the case of the UPF, the context is represented by the PFCP Session, which contains the following rules:

- Packet Detection Rules (PDRs)-Rules for packet detection;
- Forwarding Action Rules (FARs)-Rules for forwarding packets;
- QoS Enforcement Rules (QERs)-Rules for applying QoS;
- Usage Reporting Rules (URRs)-Rules for generating reports;
- Buffer Action Rules (BARs)-Rules for buffering packets;
- Multi-Access Rules (MARs)-Rules for traffic steering functionality.

Table 1 shows the relationship between the signaling interfaces (5G and LTE) used between the CP and UP of 5G and LTE networks and the rules applied in the packet. We will see that the upf-bpf supports PDRs and FARs.

**Table 1.** Relation between the signaling interfaces (5G and LTE) between the CP and UP and the packet rules applied.

| Rules | Interfaces | | | |
|---|---|---|---|---|
| | **Sxa** | **Sxb** | **Sxc** | **N4** |
| PDR | x | x | x | x |
| FAR | x | x | x | x |
| URR | x | x | x | x |
| QER | - | x | x | x |
| BAR | x | - | - | x |
| MAR | - | - | - | x |

The flow for processing packets in the UP is represented in Figure 5. When the packet is received, its header is analyzed to find the PFCP Session to which the packet belongs. Once the session is found, the UP looks up the highest precedence The PDR contains the Packet Detection Information (PDI), which has Information Elements (IEs) that will be matched with the header of the received packed. For instance, the PDI contains the UE IP address that can be source (uplink) or destination (downlink) IP address of the packet, depending on the traffic direction. After the PDI matches with the header of the packet, there might be more than one PDR that has the the same PDI IEs. However, just the highest precedence is selected. In the next step, the selected PDR contains the rules (QERs, URRs, BARs or MARs) that are applied, if they are available. Only the FAR is mandatory according to specification [17]. Finally, the packet can be forwarded to the network interface as defined in the FAR. It is important to highlight that these steps fit both 5G networks (UPF) and LTE networks (SPGWu) [17].

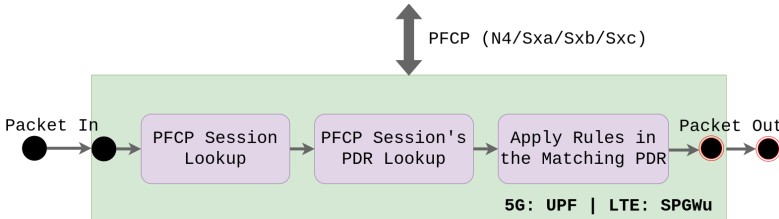

**Figure 5.** Flow of packet processing in UP function defined by UPF and SPGWu, components of NGC and EPC respectively.

One of the protocols used to load user data packets is GPRS Tunneling Protocol User Plane (GTP-U) [18]. The original packet (IP datagram) is called a Transport Protocol Data Unit (T-PDU). When combined with the GTPu header, the packet is called a GTP encapsulated user Protocol Data Unit (G-PDU). The diagram representing the protocol stack used in the communication between the UP elements is represented in Figure 6. The Tunnel Endpoint ID (TEID) indicates which tunnel a particular T-PDU belongs to and

it is part of GTP Standard Header. The GTP Extended Header is set whenever one of the flags Sequence Number, NPDU number or Extension Header is set.

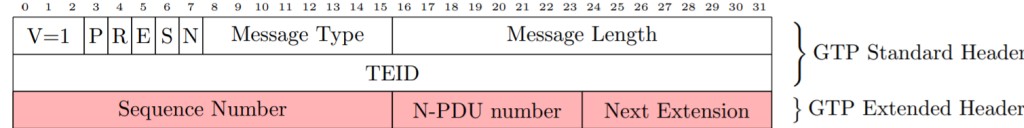

**Figure 6.** GTP-U header with extensions. Source: [19].

*2.5. The UPF-BPF Project*

The upf-bpf project is an open source C++ library powered by BPF/XDP for user planes in the mobile core network (5G/LTE). The main goal is to enable in-kernel fast packet processing in third-party UPF/5G or SPGWu/LTE components in order to: (i) boost those components which do not support fast packet processing or (ii) co-locate them with other fast packet processing solutions (e.g., DPDK). The possible scenarios that take advantage of this type of technology: MEC, 5G Non-Public Networks (NPN), on-premise and 5G enterprise. The main features supported are:

i   PFCP session management: create, read, update, and remove PFCP sessions, PDRs and FARs;

ii  Fast packet processing for uplink and downlink user data traffic: classify and forward UDP and GTP traffic based on PDR and FAR, respectively.

The data model is shown in Figure 7. The library is divided in two layers: Management Layer and Datapath Layer. The high level library design is shown in Figure 8.

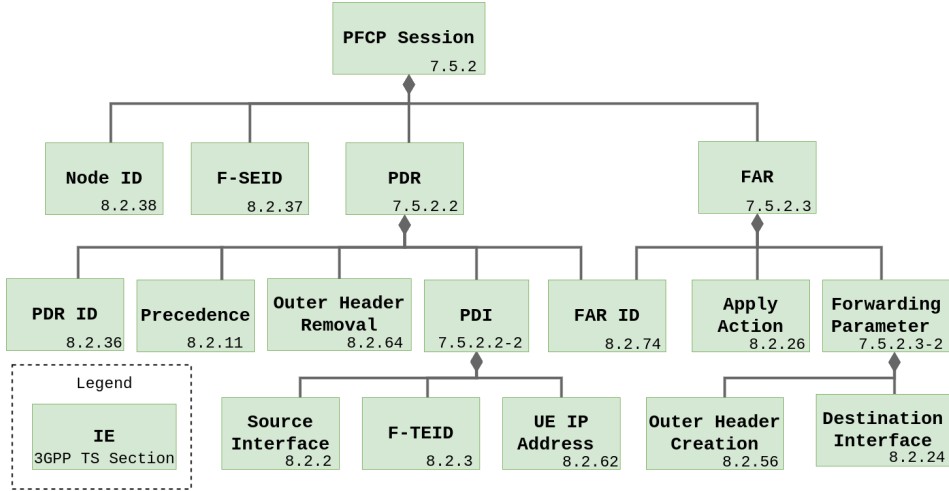

**Figure 7.** The PFCP context data model of the upf-bpf project.

Management Layer: It is a user space layer to manage PFCP sessions and BPF programs lifecycle. A client can create/read/update/delete PFCP sessions through API. When a PFCP session establishment request message is received by the user plane component, the message is parsed and a call is made to the library via PFCP Session Manager API. The PFCP Session Manager calls the BPF Program Manager to load dynamically a BPF bytecode, which represents the new PFCP session context, i.e., there is a BPF program running in kernel space for each PFCP session. The program contains the BPF maps used to store the PDRs and FARs. All the communication between the user space and the kernel space is through the libbpf library [20], which is maintained by the Linux kernel source tree. The PFCP Session Manager parses the structures received to a BPF map entries and updates the maps with them. The PFCP session context is created in Datapath Layer, where the user traffic will be handled.

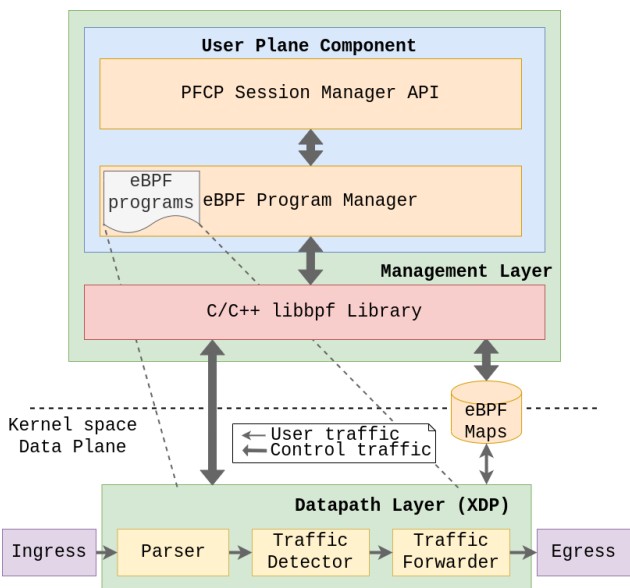

**Figure 8.** High level design of the user plane library using BPF/XDP.

Datapath Layer: It is a kernel space layer to process the user traffic inside the XDP. A service chain function was created with three main components: the Parser, the Traffic Detector and the Traffic Forwarder. The Parser parses the ingress traffic to check if it is a uplink (GTPu) or a downlink (UDP) flow. If it is an uplink/downlink flow, the TEID/UE IP address key is used to get the PFCP session context. Then, the packet is passed to the PFCP session context represented by a BPF program via tail calls. Here, the Traffic Detector accesses the BPF hash maps in order to find a PDR associated with the packet. If there is a PDR stored, the packet passes to the Forwarder. Finally, the Forwarder uses the FAR ID from the PDR to find the FAR, which is stored in a BPF hash. The FAR contains the action (e.g., forward) that will be applied, the outer header creation and the destination interface. Besides, the Forwarder accesses other BPF maps to check the MAC address of the next hop and the index of the destination interface where the packet will be redirected. The datapath workflow is shown in Figure 9.

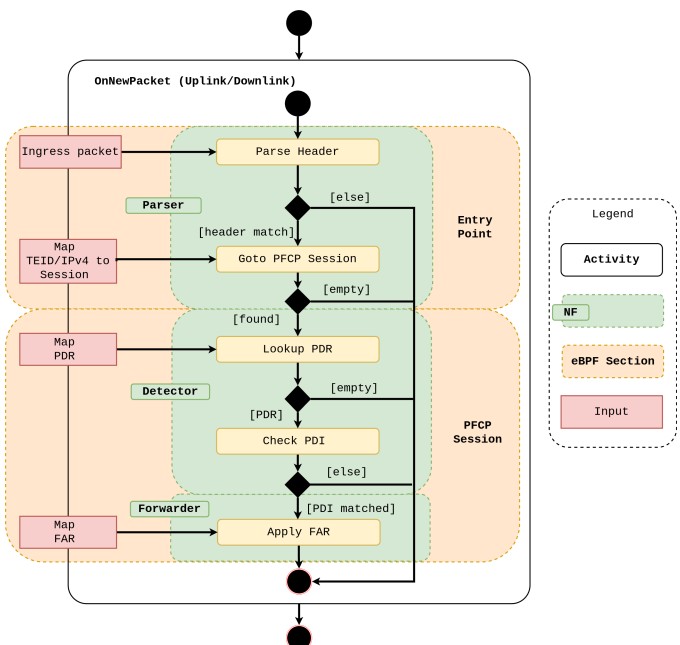

**Figure 9.** Workflow in Datapath Layer (version 1).

## 3. Current Limitations

In the version presented in [7], the main limitations can be divided into two categories: functionality and flexibility.

### 3.1. Functionality Constraints

For each TEID/UE IP address, which is the key of the BPF hash maps for uplink/downlink flow, the control plane can map only one PDR. Based on [17], there might be more than one PDR mapped to the same TEID. The PDI also may take account of other IEs to lookup the PDR, for instance, the Source Interface (e.g., Access or Core, which denote an uplink and downlink traffic direction respectively). Therefore, there could be two PDR mapped to the same TEID, but with a different source interface. However, this does not solve our problem, because it is possible to have two PDRs with the same PDI. So, how does the Datapath Layer differentiate between them? Here the IE Precedence is used to define the highest precedence between them. This solves our problem: create an interaction between the PDRs which has the PDI that matches with the header of the packet and find the highest precedence between the PDRs that were found. However, the way that was designed, all the PDRs are stored in the Datapath layer. In order to find the highest precedence, an iteration (loop) would be implemented to lookup the matched PDR inside the Datapath Layer (Figure 10). The solution does not scale, because the loop increases the size of the BPF program, leading the verifier to reject the BPF when loaded into the kernel. The verifier in Linux kernel has a limitation to verify 1M instruction maximum [21]. Besides, the latency and the time to load the BPF program increase due to the PDR lookup loop and the size of the BPF program, respectively. Both may impact the performance for 5G ultra Reliable Low-Latency Communication (uRLLC) use cases.

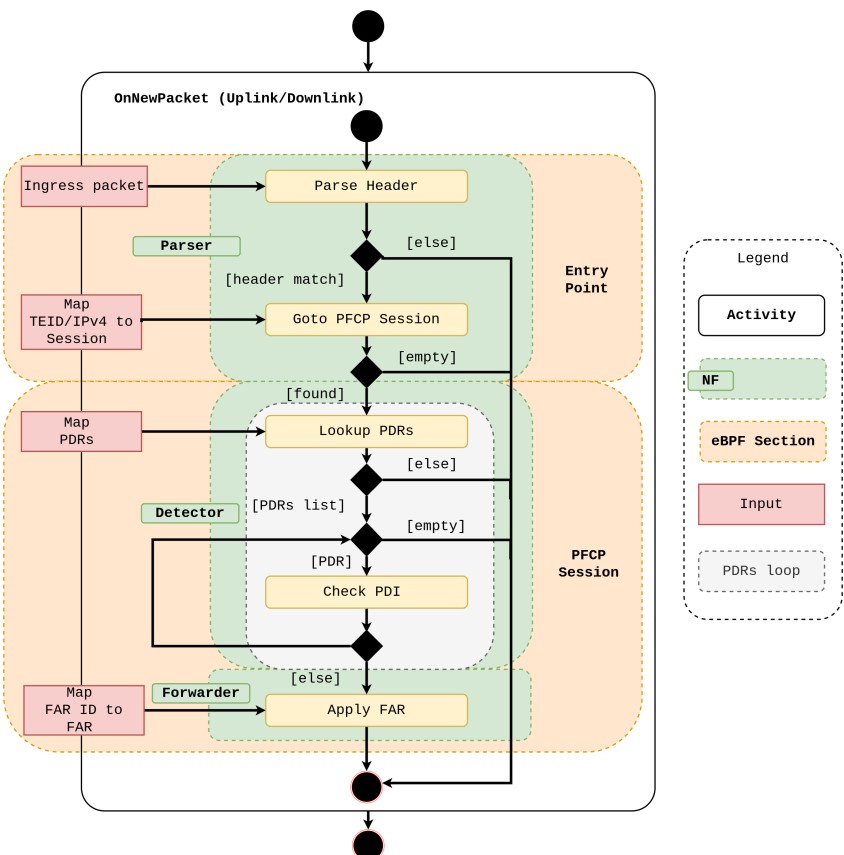

**Figure 10.** Workflow in Datapath Layer with PDR lookup loop (version 1).

### 3.2. Flexibility Constraints

Adding new rules (e.g., QER, URR, etc) in the current version involve to add a new PFCP session context (BPF section), because all the rules are coupled in the same BPF section (Figure 9). Then, the BPF program (PFCP session) becomes bigger whenever we add support for new rules. Besides, the time to load it into the kernel also increases, leading the solution to have less time to respond to changes in the network (less flexible). Furthermore, the current design does not support modifying the pipeline inside the PFCP session BPF section. Figure 11 shows the current design adding the QER support. There might be some cases that a PFCP session is composed of only one FAR (just the FAR is mandatory [17]). In this case, whenever the packets arrive, the BPF program will lookup if there is a QER and it will skip the apply QER action. This operation leads to the increase of the packet processing latency. This check could be avoided if there is one BPF program for each rule and the Management Layer only deploys those programs (rules) that are available inside the highest priority PDR of the PFCP session.

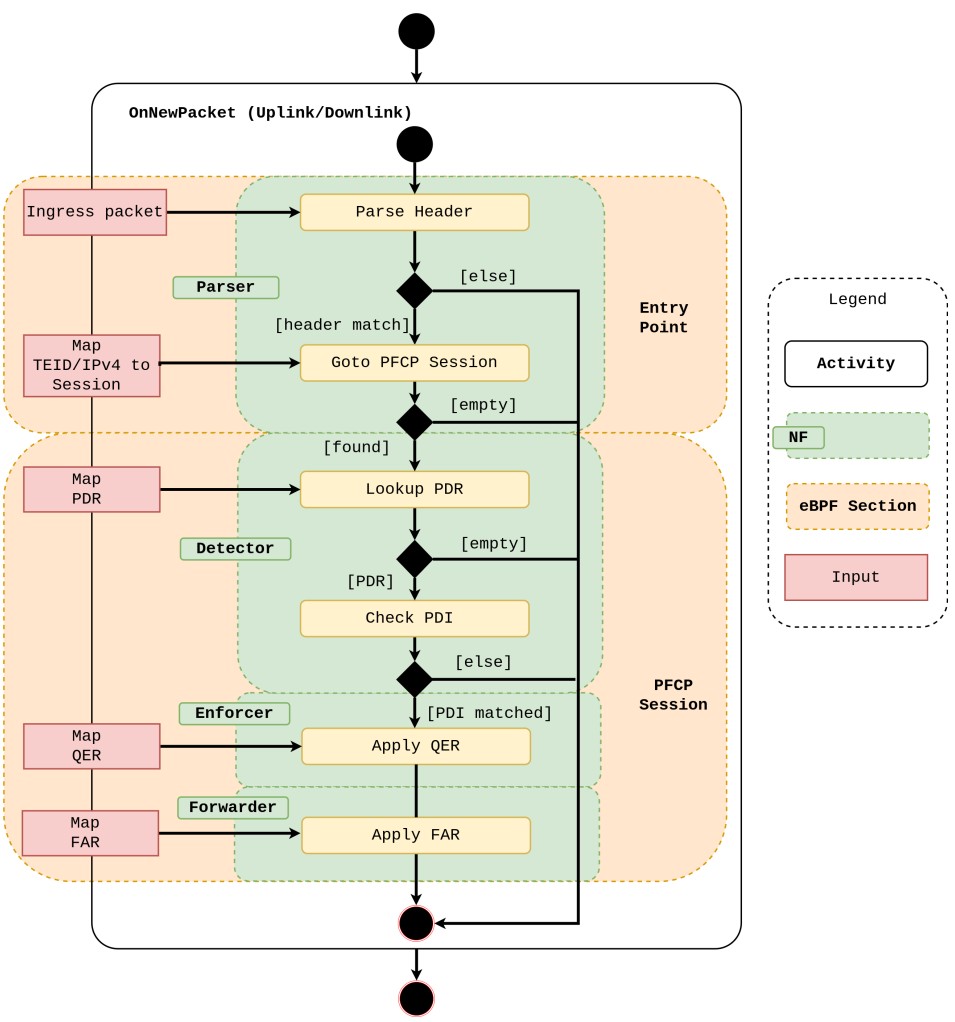

**Figure 11.** Workflow in Datapath Layer with QER and FAR (version 1).

## 4. New Design

A new design was created to fix the limitations discussed in the previous section. Here, we will explain the main differences of the design in the Management Layer and the Datapath Layer.

*Management Layer*: The main difference with the previous version is that instead of loading all the PFCP session context on the Datapath Layer, these information are stored in the Management Layer and only the highest precedence PDR along with the associated

rules for each PFCP session will be deployed in the Datapath Layer. Figure 12 shows the activity diagram of the PFCP session creation use case.

*Datapath Layer*: The main difference between the previous version is that instead of mapping one BPF program to one PFCP session, this version maps one BPF program for each rule defined in highest precedence PDR (e.g., FAR) for each PFCP session created. For instance, if one PFCP session is composed by two PDRs and the high precedence PDR is composed by one QER and one FAR and the other one is composed by one FAR, then two BPF program will be deployed on the Datapath Layer, one for the QER and another for the FAR related to the PDR with the highest precedence. The main advantage of following this approach is to avoid the PDR loop (Figure 10). Furthermore, the BPF programs are more decoupled which can be easily extensible for new rules (e.g., QER, URR). The pipeline is also more flexible, which can be changed based on the rules contained in the PFCP session. The Figure 13 shows the activity diagram for the OnNewPacket use case. In this case, the Datapath layer supports only FAR and QER. The activity diagram presented on Figure 14 is more generic. The xRule can be one of the 5G rules described in Section 2. The xRule is an abstraction of the one rule associated with the highest precedence PDR. If there are more than one rule, the xRule will be chaining.

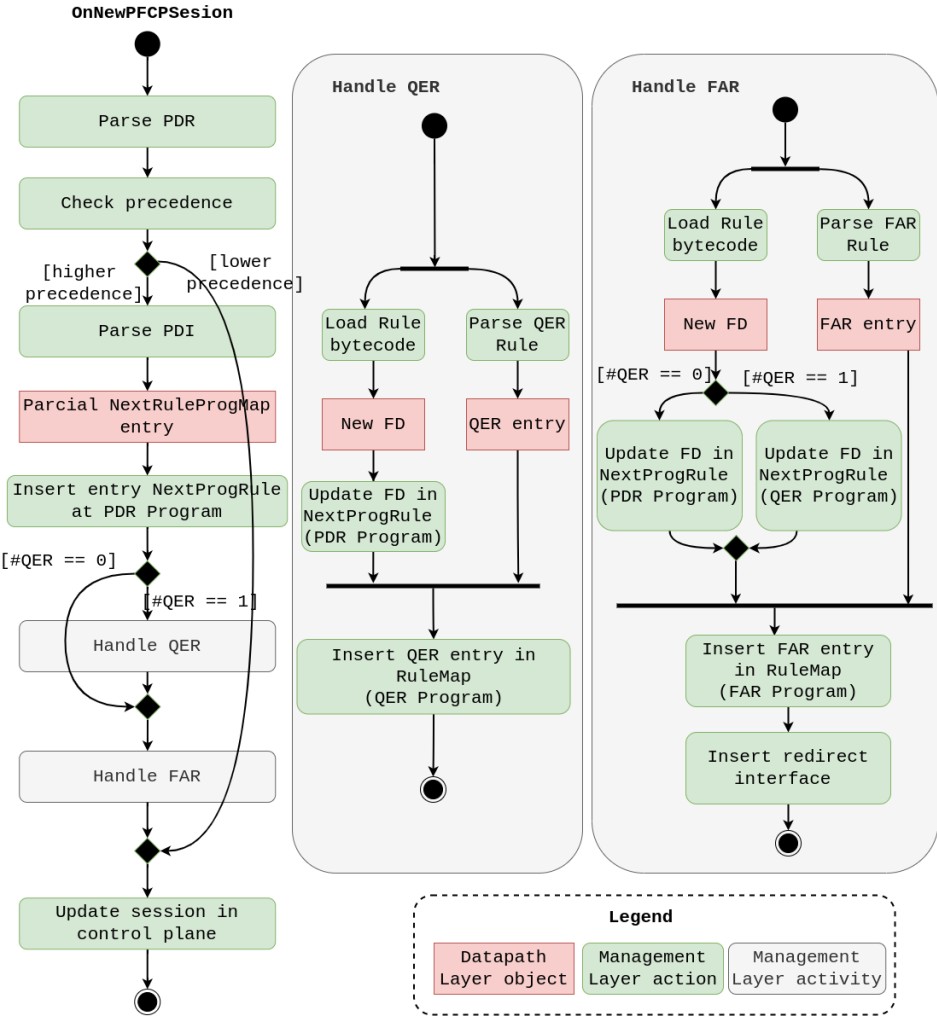

**Figure 12.** PFCP session creation activity diagram in Management Layer (version 2).

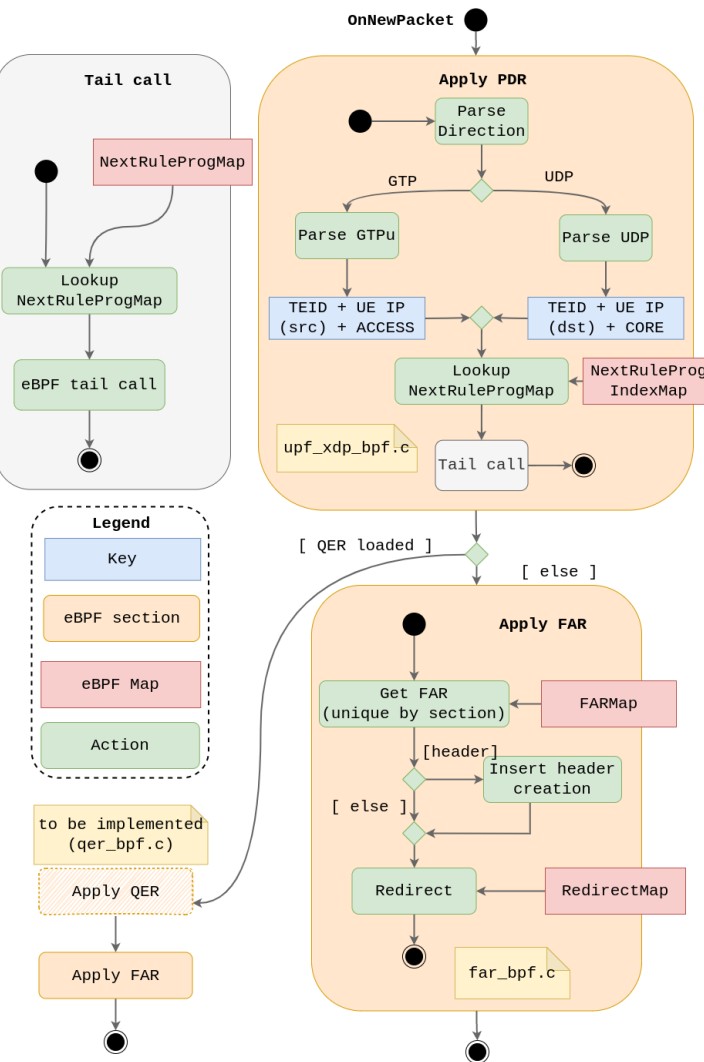

**Figure 13.** On new packet received activity diagram in Datapath Layer (version 2) .

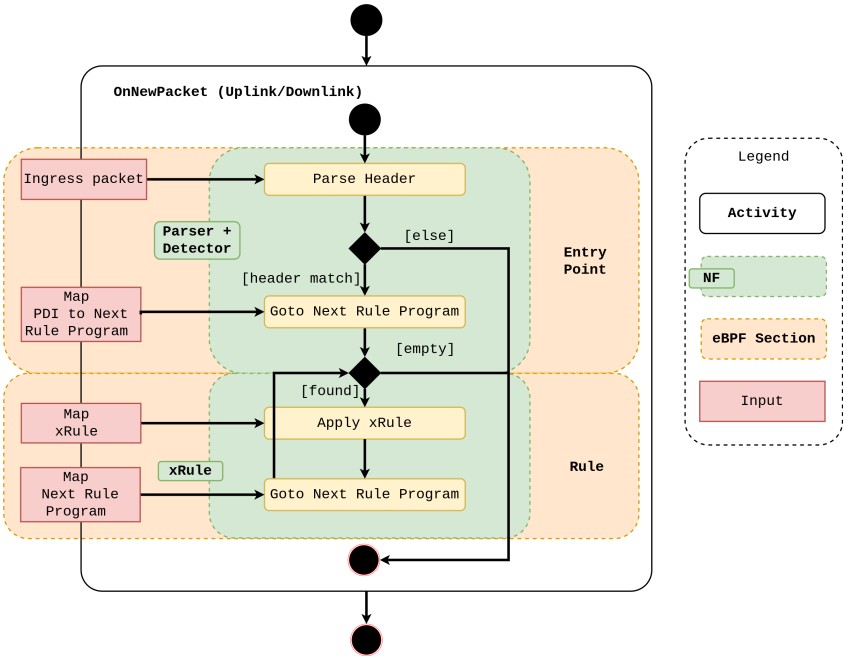

**Figure 14.** On new packet activity diagram with the decoupled rules (version 2).

## 5. Performance Evaluation

### 5.1. Setup

We evaluate the new design through RFC2544-like measurements in the same testbed setup as in our previous work [7]. It is composed by two server Intel(R) Xeon(R) CPU E5-2620 v2 @ 2.10 GHz, 32 GiB of the DRAM, 15M of L3 cache, 6 cores (hyper-threading disabled), dual-port 82599ES 10-Gigabit SFI/SFP + NIC. Both machines have Ubuntu 20.04.02 LTS installed with Linux kernel v5.4.0-72-generic compiled with BTF flags enabled. One machine is used to generate user traffic with TRex Traffic Generator [22] and the other is the DUT (Device Under Test) where the proposed solution is deployed. The NICs were configured with the toeplitz hash algorithm. The setup is shown in Figure 15.

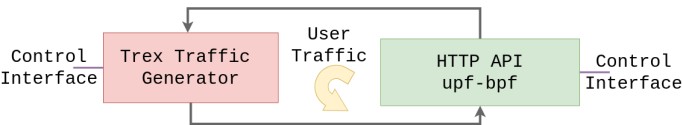

**Figure 15.** Testbed setup.

### 5.2. Scalability

We run the following experiment to evaluate the scalability and the workload when the solution achieves the maximum throughput for a specific traffic generation flow. Both uplink and downlink scenarios were tested. The tests consisted in (i) run Trex Traffic Generator server, (ii) run HTTP API (DUT), (iii) send a PFCP session establishment request via HTTP API to DUT, (iv) configure the number of Rx queue [23] in DUT, (v) generate traffic (GTP or UDP) using Trex Traffic Stateless API and, finally, (vi) collect CPU utilization per core (DUT) and throughput (Trex Traffic Generator) metrics. We create a JSON message to define the PFCP session establishment request to be sent to DUT. This message contains the PDRs (uplink and downlink) and FARs and the static ARP table of the next hop. The PFCP session context message is shown in Listing 1.

The traffic was generated using the Field Engine modules available in TRex Stateless API [24]. This engine generates 1000 flows varying the source IP address randomly. This technique is used to avoid the assignment to a specific receive queue, distributing the traffic between the receive queues in the DUT. So, this improves the throughput due to the load balance between the cores. The GTP or UDP headers are generated depending on the test (uplink or downlink) based on the PFCP Session Establishment Request message. In both cases, the frame size is 64 bytes.

### 5.3. Adaptation Time

In this experiment, we measure the time spent to inject (load) the BPF program when a new PFCP session is created after receiving the PFCP Session Establishment Request message (Listing 1). In version 1, the time is related to the PFCP session BPF section (Figure 9) injection. In version 2 (proposed solution), it is related to the Rule BPF section (Figure 13) injection. The rule in this case is the FAR of the highest precedence PDR (FAR ID equal to 100). The time was measured using the LOG_FUNC (https://github.com/navarrothiago/upf-bpf/blob/a6 45eb16742f88f155770d5bd8146f0edd6431ce/src/utils/LogDefines.h#L8 accessed on 18 March 2022) macro, which record the time when the execution enter and exit the scope of load function (https://github.com/navarrothiago/upf-bpf/blob/a645eb16742f88f155770d5bd814 6f0edd6431ce/src/programs/ProgramLifeCycle.hpp#L143-L157 accessed on 18 March 2022).

### 5.4. Results

Figures 16 and 17 show the throughput per number of cores and the distribution of the CPU utilization, respectively. We observe a clear linear relationship between the number of cores and the throughput. The implemented solution scales with the number of the cores, achieving 10 Mpps for downlink and 11 Mpps for uplink. This difference is related to the actions performed in each direction. For the downlink, the UPF encapsulates the GTP

header. The packet size is greater than 64B after reception in the TRex Traffic Generator. On the other hand, for the uplink, the UPF decapsulates the GTP header and the packet size becomes smaller than 64B. These numbers reflect the packet forwarding performance with packet loss less than 0.5%. Besides, almost 30% of the CPU is idle, which could be used by other tasks when 6 cores are used. We highlight that our results are aligned to those presented in [7]. Therefore, our flexible design does not introduce performance hits.

**Listing 1.** The JSON message representing the PFCP Session Establishment Request used in the tests.

```
{
  "seid": 1,
  "pdrs": [ {
      "pdrId": 20,
      "farId": 200,
      "outerHeaderRemoval": "UDP_IPV4",
      "pdi": {
        "teid": 0,
        "sourceInterface": "INTERFACE_VALUE_CORE",
        "ueIPAddress": "10.1.3.27"
      },
      "precedence": 2
    }, {
      "pdrId": 10,
      "farId": 100,
      "outerHeaderRemoval": "GTPU_UDP_IPV4",
      "pdi": {
        "teid": 100,
        "sourceInterface": "INTERFACE_VALUE_ACCESS",
        "ueIPAddress": "10.1.3.27"
      },
      "precedence": 1
    } ],
  "fars": [ {
      "farId": 200,
      "forward": true,
      "forwardingParameters": {
        "outerHeaderCreation": {
          "outerHeaderCreationDescription": "GTPU_UDP_IPV4",
          "ipv4Address": "10.1.3.27",
          "portNumber": 1234
        },
        "destinationInterface": "INTERFACE_VALUE_ACCESS"
    } }, {
      "farId": 100,
      "forward": true,
      "forwardingParameters": {
        "outerHeaderCreation": {
          "outerHeaderCreationDescription": "UDP_IPV4",
          "ipv4Address": "10.1.3.27",
          "portNumber": 1234
        },
        "destinationInterface": "INTERFACE_VALUE_CORE"
    } } ],
  "arpTable": [ {
      "ip": "10.1.2.27",
      "mac": "90:e2:ba:27:fd:3c"
    }, {
      "ip": "10.1.3.27",
      "mac": "90:e2:ba:27:fd:3d"
    } ] }
```

Regarding the higher load in the core #1 and #2 in Figure 17, the main reason could be related to the load balancing between the cores. Basically, four factors contribute to the load balancing: NIC hash algorithm, NIC hash key (tuple), NIC indirection table and the flows. Each core is associated with one receive queue. The core processes only the packets from the queue that was assigned to it. So, packets can be pushed more to some specific core, leading to an asymmetric load between the cores. This analysis is not the scope of this article.

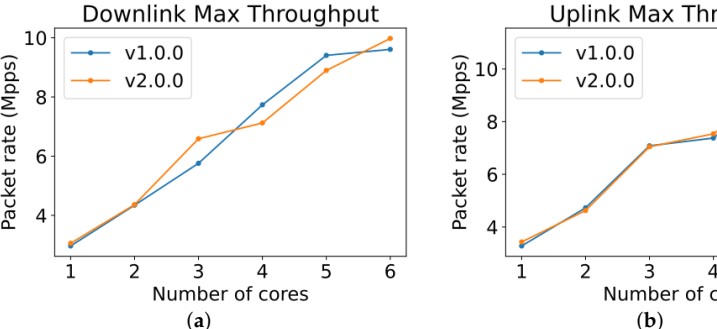

**Figure 16.** Scalability of the proposed solution. The solution achieves almost 10 Mpps for downlink and 11 Mpps for uplink with packet loss less than 0.5%. (**a**) Downlink. (**b**) Uplink.

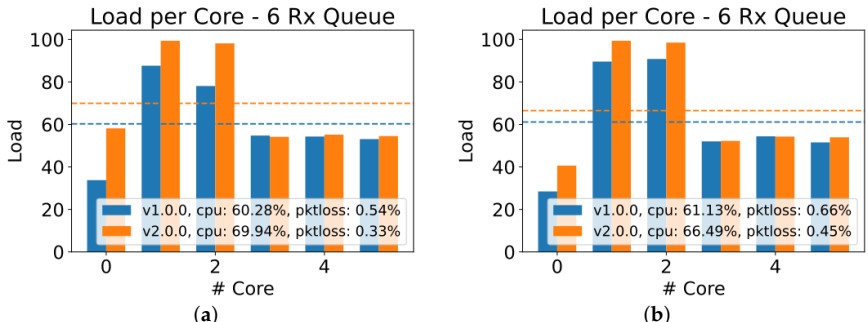

**Figure 17.** CPU utilization when varying number of cores. The cores #1 and #2 are with a higher CPU utilization. (**a**) Downlink 6 cores. (**b**) Uplink 6 cores.

The time spent to inject the BPF program for each version is shown in Table 2. For version 1 (1.0.0), it costs 27 ms to inject the BPF program after receiving a PFCP Establishment Request message, which is composed with one FAR. On the other hand, for version 2 (2.0.0), only 1 ms (time reduction of 96%). Besides, the number of instructions decreased by 32%. The main reason is due the logic related to lookup the PDR is implemented in the control plane (Management Layer). The results follow the correlation between the number of instructions and the time to inject the BPF program into the kernel presented in [10].

**Table 2.** Time spent to inject BPF program into the Linux kernel (JIT compiler phase) after receiving a PFCP Establishment Request message with only one FAR.

| Version | BPF Section | BPF Insn | Injection (ms) |
|---------|-------------|----------|----------------|
| v1.0.0 | PFCP Session | 402 | 27 |
| v2.0.0 | FAR | 272 | 1 |

## 6. Related Work

Since the introduction of concepts such as NFV, several works have been carried out addressing the issue of fast packet processing [25] in general, and tailored to 5G [26].

The authors in [27] present a prototype for packet processing based on hardware using the programmable platform NetFPGA with the programmable data plane in the P4 language. The work consists of developing a firewall located between the core and the edge of 5G networks. The firewall is responsible for analyzing the internal IP headers of each packet, unlike the traditional firewall, which only analyzes the external header. In parallel, [28] also proposes a similar solution, but with a focus on multi-tenancy scenarios to ensure quality of service for applications with strict latency requirements, with tests performed with OpenAirInterface (OAI) [29]. Although FPGA-based solutions perform well, they are considered expensive compared to general purpose CPU as well as complex

to develop due to their low-level hardware description languages (HDL) such as the VHDL language [25]. It is important to point out that these presented solutions do not seem to be ideal to be implemented in datacenters infrastructures located at the edge, which have restrictions regarding the implementation cost [30].

Regarding software-based packet processing solutions, [31] presents a prototype using the modular router, Click [32], integrated with the framework Netmap [33] for transferring session context between base stations. In addition, [34] evaluates a DPDK-based [35] prototype for a media gateway located in the IP Multimedia Subsystem (IMS). DPDK-based solution can increase performance 10 times for packet processing [36], however it is surpassed by XDP technology in scenarios when packet forwarding happens through the same interface [8].

To the best of our knowledge, [37] is the main publicly available work closest to our `upf-bpf` project [37] presents a prototype of a gateway based in BPF using TC and XDP technology for fast packet processing with a focus on developing a component that can be deployed on the edge. Developed inside the Polycube Framework [38], the main features presented are packet forwarding, classification and QoS policies. It was The main advantages of our work comparing with [37] are (i) it is decoupled to a specific frameworks, (ii) it is aligned with 3GPP specifications, and (iii) it is based on `libbpf` instead of BCC [39], which does not belong to Linux source tree and depends on the `clang` run-time compiler. Therefore, we argue that our solution can be easily integrated with different software-based 5G uses plane solutions.

As far as we understand, none of the related work explores run-time flexibility in UPF as carried out in our work. On the P4 programmable data plane front, efforts are being devoted (e.g., FlexCore [40]) to make networks run-time programmable, totally aligned with our flexibility goals for `upf-bpf`.

The approach presented in this paper may leverage open source projects for telecommunication networks core, such as srsLTE [41], OAI [29], Open5GS [42], UPF-EPC [43], Magma Facebook [44], and Free5Gc [45]. Only srsLTE [41] does not provide support for NCG functionalities. The OAI, srsLTE, and Open5GS solutions do not have specific technologies for fast processing in the UP, which has been developed in the operating system's user space. Already UPF-EPC, Magma Facebook, and Free5Gc present kernel-level UP implementations using the kernel module gtp5g [46], BESS (Berkeley Extensible Software Switch) [47] and OvS (Open vSwitch) [48], respectively. With regard to CUPS support, we can highlight OAI, UPF-EPC, Open5GS, Free5Gc, and Magma Facebook. Although the Magma Facebook is based on an older version of the OAI without CUPS support, the solution was built using the SDN Ryu controller [49] and OvS. We believe all these projects could leverage our proposed solution, especially those that do not support fast packet processing, like OAI, srsLTE, and Open5GS. The summary of related work on fast packet processing and 5G networks is provided in Table 3.

**Table 3.** Related work. Legend: U/D-Under Development; N/A-Not Apply; All-All NCG components; U-Unavailable.

| Academic Research | | | | | | | |
|---|---|---|---|---|---|---|---|
| | 5GS | | Fast Packet Processing | | Experimental Evaluation | | |
| **Work** | **Location** | **Component** | **Based on** | **Technology** | **Environment** | **OSS CN** | **Application** |
| [37] | Edge | UPF | SW | BPF, TC, XDP | Polycube | No | QoS, Traffic forward |
| [34] | N/A | N/A | SW | DPDK | Standalone | No | 5G Media Gateway |
| [31] | Core | UPF | SW | Netmap, Click | Docker Container | No | Context migration, service chain |
| [28] | Edge & Core | gNB & UPF | HW | NetFPGA, P4 | OAI | Yes | Multi-Tenancy, QoS, Multimedia |
| [27] | Edge & Core | gNB & UPF | HW | NetFPGA, P4 | Standalone | No | Firewall |
| This Work | Edge | UPF | SW | BPF, XDP | Standalone | Yes | Traffic forward |

**Table 3.** *Cont.*

| | | | | | | | |
|---|---|---|---|---|---|---|---|
| **Open Source Software** | | | | | | | |
| | **5GS** | | **Fast Packet Processing** | | | **Details** | |
| **OSS** | **Location** | **Components** | **Based on** | **Technology** | **CUPS** | **Language** | **Application** |
| [29] | Core | U/D | U | U | Yes | C/C++ | CN SA/NSA |
| [41] | Core | U | U | U | No | C++ | CN NSA |
| [44] | Core | U/D | U | OvS, Kernel module | No | C | CN NSA |
| [43] | Core | UPF | SW | BESS | Yes | C++ | CN SA |
| [42] | Core | U/D | U | U | Yes | C | CN SA |
| [45] | Core | All | SW | Kernel module | Yes | Go/C | CN SA/NSA |
| This Work (upf-bpf) | Core | UPF | SW | BPF, XDP | Yes | C++ | CN SA/NSA |

## 7. Conclusions and Future Work

This work addressed the current limitations of the `upf-bpf` project and presented a new design in order to improve the flexibility by reducing the time to inject the BPF program into the Linux kernel in run-time after receiving a PFCP Establishment Request message. Our evaluation showes that the new solution can achieve high throughput without consuming all the CPU resources at a lower run-time adaptation time. Besides, we are confident that the new design can be a reference point for those who want to create BPF-based network function implementations.

As future work, we intend to include QER in order to demonstrate user plan flexibility when new rules are created or removed and evaluate the behavior when varying the number of PFCP sessions and packet size. Furthermore, we plan to use real data traffic for tests and conduct a proof of concept to demonstrate an integration with the open-source OpenAirInterface (OAI) framework. In addition, we plan to automate and standardize the input and output of the performance benchmarking experiments following the Gym methodology [50] and its open source framework [51].

**Author Contributions:** Conceptualization, T.A.N.d.A., R.V.R., D.F.C.M. and C.E.R.; methodology, T.A.N.d.A., R.V.R.; software, T.A.N.d.A.; validation, T.A.N.d.A.; investigation, T.A.N.d.A., R.V.R.; writing—original draft preparation, T.A.N.d.A.; writing—review and editing, T.A.N.d.A., R.V.R., D.F.C.M. and C.E.R.; supervision, R.V.R., D.F.C.M. and C.E.R.; project administration, R.V.R. and C.E.R.; funding acquisition, C.E.R. All authors have read and agreed to the published version of the manuscript.

**Funding:** This work was supported by grants 2018/23101-0 and 2020/05182-3 from the São Paulo Research Foundation (FAPESP).

**Data Availability Statement:** Publicly available datasets were analyzed in this study. This data can be found here: https://github.com/navarrothiago/upf-bpf (accessed on 18 March 2022).

**Conflicts of Interest:** The authors declare no conflict of interest.

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
