# Peer review of "Run-Time Adaptive In-Kernel BPF/XDP Solution for 5G UPF"

_electronics, doi:10.3390/electronics11071022_

Round 1

Reviewer 1 Report

This paper presents an in-kernel solution based on BPF and XDP to implement the UPF of 5G. The proposed solution is an evolution of the original proposal, named upf-bpf. They evaluate a new design to improve its flexibility and surpass some of the limitations that implies the use of BPF, such as a limited number of instructions. The results show a significant improvement in the injection of BPF programs into the kernel with respect to the original one, while keeping the same functionality and not suffering throughput penalties. The main contribution of this work is the proposal of this component as a reference design to develop run-time adaptive BPF solutions for the 5G UPF, with the intention of testing its integration in 5G core open-source solutions such as OAI.

The article is very well written, it is easy to follow the main ideas of the work and to understand the proposal of the authors. The english style is correct and only a couple of typos have been found. Besides, the contribution is clear and the methodology and results are perfectly explained and defined.

With regard to the contribution itself, the idea is very promising and it is well executed. Exploiting the advantages of BPF and XDP to deliver adaptive and dynamic UPF operation is in line with the current worldwide research lines focused on the importance of data plane reprogramability.

Only some typos/suggestions to be corrected:
 - Lines 137-8: "After the PDI matches with the header of the packet, it might have more than one PDR the conditions." Please, rewrite the last part of the sentence to make it clearer to the reader.
 - Lines 170 and 175: "an BPF". a BPF.
 - Figure 6: I suggest to move the figure legend to the top right corner to improve the readability.

In summary, it is the reviewers opinion that the work is acceptable for publication with those minor corrections. I believe that the solution is relevant for the network reprogramability paradigm, which is a fundamental pillar of 5G and future networks.

Author Response

We sincerely thank Reviewer 1 for the valuable comments/suggestions that helped us to improve our manuscript. We have revised the manuscript accordingly, as noted in all changes in the revised manuscript coloured blue for new additions/edits. 

- [x] Lines 137-8: "After the PDI matches with the header of the packet, it might have more than one PDR the conditions." Please, rewrite the last part of the sentence to make it clearer to the reader.
- [x] Lines 170 and 175: "an BPF". a BPF.
- [x] Figure 6: I suggest to move the figure legend to the top right corner to improve the readability 

Reviewer 2 Report

The present work describes an incremental improvement over the authors' previous work on a "BPF/XDP Solution for 5G UPF". The authors alter their previous design to increase run-time flexibility and significantly decrease BPF program injection time into the kernel. From a performance perspective no noticable changes are observed in their measurements. Using BPF/XDP for a software implementation of the UPF at least partially in kernel space is a timely topic in light of the current trend towards cloud-based 5G Core and indeed even partial RAN implementations. The presented improvement in BPF injection time is certainly noteworthy but I am not entirely convinced of the significance beyond the upf-bpf project. For a journal publication of an otherwise unchanged paper, I would like to see a bit more in terms of evaluation. The authors cite multiple alternative implementations such as Free5GC, OAI and srsRAN which offer their own implementations of a 5G UPF or at least an LTE S/PGW. In addition to the architectural changes, I'd like to see this version compared in terms of performance to at least one alternative implementation to answer the question, if indeed the BPF/XDP implementation is the best path forward. Without at least that information, it is hard to judge whether the change in architecture is actually worthwhile and should be published in its present form of just be left to read on GitHub.

Author Response

We sincerely thank Reviewer 2 for the valuable comments/suggestions that helped us to improve our manuscript. We have revised the manuscript accordingly. All changes in the revised manuscript are colored blue to highlight new additions/edits.

The main project objective is to provide a library to enable fast packet processing for the UPF/SPGW. According to the presented results, we believe that the proposed approach would perform better than OAI and srsRAN, for instance, which are based on Linux sockets. We agree that would be good to compare with Free5GC, which is based on the gtp5g kernel module. The publication focuses firstly to fill critical flexibility aspects missing in the first version of the architecture. We present a new design in order to overcome relevant issues. In future work, we will compare with other open source solutions soon. We believe that the lessons learned and the open-source code contributions of the new design will be valuable and useful to the broader community. Indeed, we intend to keep updating the github repository of the project as it keeps evolving but we believe that a journal publication would contribute to the overall visibility and scientific impact of the work.

Reviewer 3 Report

Abstract: Overall, I am missing a piece of information in the abstract that lets the reader know what the authors actually did, i.e., how they improved actually the upf-bpf software.

Introduction: Similar to the abstract, I am actually missing a piece of information that explains in the introduction (1) what the challenge is and (2) what the authors actually did/contributed to address the existing challenge.

Background: Overall, I liked the background section, but believe that it could still be polished. This would be great!

- An explanation of how BPF and XDP interact would be good.

- Maybe a table with all acronyms would help.

- There is, unfortunately, no clear explanation of the Sxa/Sxb/Sxc interfaces.

- Table 1 and its content is not really clear as the information about the interfaces, in my point of view, is not clear.

- For the UPF, I do not fully understand the point about the priorities. Maybe this due to my lack of knowledge about the 5G architecture in general, but what happens if there are rules with the same priorities? Why should there be rules with different priorities? In particular, I believe that this aspect is important as this part is actually exploited to improve the performance of the software. Please elaborate more on the priorities and their purpose, also within the 5G architecture context.

- Generally, many figures carry a lot of information, however, they are not really well-explained in the text. This should be improved. Also the captions could deserve some better explanations of the figures. This would help the reader.

- Generally, for a non BPF/XDP expert, it is a bit hard to follow which data structure is used/deployed where and how the interaction between both, bpf and xdp happen.

Current Limitations:

- Maybe having Figure 9 and 10 next to each other would help to see the differences.

- As understood from the text, is it correct that you exploit the idea to load only the rule with the highest priority as BPF programs? So, why can there be PDRs with different precedence then? How can this happen?

- So, it is ok that the datapath layer supports only FAR and QER, or is this specific to the example given here?

Performance Evaluation:

- Is Listing 1 so important?

- How many runs were conduct? Why are there no statistical measures in Fig. 15 and 16? This should be improved, please.

- In Fig. 16, why are Core 1 and Core 2 highly loaded whereas the others not? Where is the different behavior here coming from?

Language:

ll 108: in the Figure 3 → in Figure 3

ll 135 be match with → be matched with

ll 138 it might have more than one PDR the conditions → broken sentence

ll 142 these steps fits → these steps fit?!

ll 144 the protocols used to load → meaning of sentence?!

ll 156 do not have support → do not support?

ll 156 Management Layer, → Management Layer and …

ll 166 It is an → It is a user …

ll 247 for the on new → for the new packet ?

ll 249 in section 2 → in Section 2

ll 301 BPF program after receive a → after receiving a PCFP …?

ll 317 FPGA based → FPGA-based

Author Response

We sincerely thank Reviewer 3 for the valuable comments/suggestions that helped us to improve our manuscript. We have revised the manuscript accordingly. All changes in the revised manuscript are coloured blue to highlight new additions/edits.

Abstract: 
- [X] Overall, I am missing a piece of information in the abstract that lets the reader know what the authors actually did, i.e., how they improved actually the upf-bpf software.

R: In the abstract we highlight the main contribution around the flexible design of the presented work as follows: “In this article, we evolve the upf-bpf open-source project by proposing a new design to improve its flexibility by reducing the run-time adaptation time.”

Introduction: 

- [ ] Similar to the abstract, I am actually missing a piece of information that explains in the introduction (1) what the challenge is and (2) what the authors actually did/contributed to address the existing challenge.

We agree that the challenge may not be clear especially for not practitioners of eBPF. Maybe flexibility requirements for multiple use cases like uRLLC where low response times are critical should be introduced in the text for context and motivation.

Background: 
- [x] Overall, I liked the background section, but believe that it could still be polished. This would be great!
- [x] An explanation of how BPF and XDP interact would be good.
- [ ] Maybe a table with all acronyms would help. 
- [x] There is, unfortunately, no clear explanation of the Sxa/Sxb/Sxc interfaces.
- [x] Table 1 and its content is not really clear as the information about the interfaces, in my point of view, is not clear.
- [ ] For the UPF, I do not fully understand the point about the priorities. Maybe this due to my lack of knowledge about the 5G architecture in general, but what happens if there are rules with the same priorities? Why should there be rules with different priorities? In particular, I believe that this aspect is important as this part is actually exploited to improve the performance of the software. Please elaborate more on the priorities and their purpose, also within the 5G architecture context.

From 3GPP TS 29.244 version 16.5.0 Release 16, 5.2 Packet Forwarding Model section:

On receipt of a user plane packet, the UP function shall perform a lookup of the provisioned PDRs and: 
- identify first the PFCP session to which the packet corresponds; and
- find the first PDR matching the incoming packet, among all the PDRs provisioned for this PFCP session, starting with the PDRs with the highest precedence and continuing then with PDRs in decreasing order of precedence. Only the highest precedence PDR matching the packet shall be selected, i.e. the UP function shall stop the PDRs lookup once a matching PDR is found

The specification does not give a lot of details about this information. We have changed in the text the terminology from priority to precedence to avoid any confusion.

- [ ] Generally, many figures carry a lot of information, however, they are not really well-explained in the text. This should be improved. Also the captions could deserve some better explanations of the figures. This would help the reader. - Generally, for a non BPF/XDP expert, it is a bit hard to follow which data structure is used/deployed where and how the interaction between both, bpf and xdp happen.

Current Limitations:
- [ ] Maybe having Figure 9 and 10 next to each other would help to see the differences.

We should think of a way to make the diagrams closer to each other.

- [ ] As understood from the text, is it correct that you exploit the idea to load only the rule with the highest priority as BPF programs? So, why can there be PDRs with different precedence then? How can this happen? 

Good question! From 3GPP TS 29.244 version 16.5.0 Release 16, 5.2 Packet Forwarding Model section:

On receipt of a user plane packet, the UP function shall perform a lookup of the provisioned PDRs and: 
- identify first the PFCP session to which the packet corresponds; and
- find the first PDR matching the incoming packet, among all the PDRs provisioned for this PFCP session, starting with the PDRs with the highest precedence and continuing then with PDRs in decreasing order of precedence. Only the highest precedence PDR matching the packet shall be selected, i.e. the UP function shall stop the PDRs lookup once a matching PDR is found

For us, this is still not clear about the usage of the precedence. However,  we believe that could be related to the Activation and Deactivation of Pre-defined PDRs. From the 3GPP TS 29.244 version 16.5.0 Release 16, section 5.19 :

To reduce the signaling overhead for establishing a PFCP session (for a PDU session or a PDN connection) and improve the signaling efficiency, the CP and UP functions may support the Activation and Deactivation of a Pre- defined PDR (ADPDP) feature…

For instance, imagine that the CP sends to UP a message to deactivate the highest precedence PDR. So, the second highest precedence PDR that matched with the incoming packet would be selected. If the  CP sends to UP a message to activate again the highest precedence PDR, it will restore the previous state. So, the precedence is useful in these scenarios. There might be other ways to handle this, but this is what was specified. 

In the Annex D (Normative): Use of PFCP over N16a for the support of traffic offload by UPF controlled by I-SMF, there is also a use case when the precedence matters. It is related to an addition of PSA and UL CL/BP controlled by I-SMF. 

- [ ] So, it is ok that the datapath layer supports only FAR and QER, or is this specific to the example given here?

The Datapath layer does not support QER. It is an example given to explain the new design.

:red_circle: Performance Evaluation:
- [ ] Is Listing 1 so important?

It is good to illustrate the PFCP contexts used during the tests.

- [ ] How many runs were conduct? Why are there no statistical measures in Fig. 15 and 16? This should be improved, please.

In the previous results  (v1.0.0) we did not present with statistical measures that is why we tried to keep the same way to present the results to make the comparison easier. 

The graph below shows the box plot for 5 executions. 

- [x] In Fig. 16, why are Core 1 and Core 2 highly loaded whereas the others not? Where is the different behavior here coming from?

Language:
- [x] ll 108: in the Figure 3 → in Figure 3
- [x] ll 135 be match with → be matched with
- [x] ll 138 it might have more than one PDR the conditions → broken sentence
- [x] ll 142 these steps fits → these steps fit?!
- [ ] ll 144 the protocols used to load → meaning of sentence?!
- [x] ll 156 do not have support → do not support?
- [x] ll 156 Management Layer, → Management Layer and …
- [x] ll 166 It is an → It is a user …
- [ ] ll 247 for the on new → for the new packet ?
- [x] ll 249 in section 2 → in Section 2
- [ ] ll 301 BPF program after receive a → after receiving a PCFP …?
- [x] ll 317 FPGA based → FPGA-based

Round 2

Reviewer 2 Report

Even thought, I believe the authors can and will do more in this area - especially in the area of performance comparisons, I believe that this article is now in a sufficiently complete state to be of interested to readers invested in the topic of software-based 5G UPFs. The eBPF approach is certainly worth publishing - even if it is only a stopgap - for now. Therefore, I have no objection to publishing this article in its current state.